# Empathy and Post-Traumatic Growth among Chinese Community Workers during the COVID-19 Pandemic: Roles of Self-Disclosure and Social Support

**DOI:** 10.3390/ijerph192315739

**Published:** 2022-11-26

**Authors:** Jinhua Dou, Chang Liu, Ruoyu Xiong, Hongguang Zhou, Guohua Lu, Liping Jia

**Affiliations:** 1School of Public Health, Weifang Medical University, Weifang 261053, China; 2School of Psychology, Weifang Medical University, Weifang 261053, China; 3Second Department of Children and Adolescents’ Psychological Behavior, Shandong Mental Health Center, Jinan 250014, China

**Keywords:** empathy, post-traumatic growth, self-disclosure, social support, community workers, COVID-19, China

## Abstract

Given the prolonged nature of the COVID-19 pandemic and its long-term psychological impacts, this study aimed to explore how empathy leads to post-traumatic growth (PTG) among Chinese community workers. Guided by the revised PTG model, this study identified the relation between empathy and PTG using a multiple mediation model that included self-disclosure and social support as hypothesized mediators. This study utilized data from 414 Chinese adults aged 20 years or older who completed an online survey during the pandemic. Self-disclosure and social support were measured as mediating variables. The study variables were positively correlated with PTG. Empathy was positively correlated with self-disclosure and social support. After controlling for demographic covariates, the results indicated that self-disclosure and social support mediated the link between empathy and PTG in both parallel and sequential fashion. Empathy, self-disclosure, and social support played important roles in the growth of Chinese community workers. The present findings have been useful in increasing our understanding, policy programs, and interventions by governments or regional bodies to ameliorate community workers’ PTG.

## 1. Introduction

Community workers are important participants in community governance in China and play the role of the main executor of community governance policies. According to statistics, since the outbreak of COVID-19, nearly 4 million community workers have been working hard on the front line of prevention and control, protecting the lives, health, and safety of urban and rural communities in all regions with professional, practical, and considerate services [1]. They stood their posts and performed their duties during the pandemic prevention and control period, playing an important role in curbing the spread of the pandemic.

The ongoing COVID-19 pandemic has placed the world under unprecedented and highly uncertain circumstances. The pandemic has disrupted the lives of individuals worldwide and caused significant psychological stress among those affected [2]. For example, community workers have been expected to work outside their normal working hours according to the need for epidemic prevention and control, including epidemic monitoring of community residents, sample collection and testing, and coordination. The dynamic nature of the COVID-19 pandemic, sudden lengthening of working hours, sudden increase in work intensity, and complexity of administrative processes have led to an increase in the psychological pressure on community workers and the prevalence of negative emotions [3]. These emotional experiences during the COVID-19 pandemic have been associated with mental health problems (e.g., anxiety, depression, and post-traumatic stress disorder) across various populations [4]. However, in the context of the pandemic, not all people have developed mental health problems, and some have even thrived further. Tedeschi and Calhoun [5] called such thriving “post-traumatic growth” (PTG) and defined it as “positive psychological change occurred as a result of struggling with highly challenging and stressful life crises (p. 1).”

Since the COVID-19 pandemic, studies on PTG have focused on health care workers [6], college students [7], and other groups [8]. Meanwhile, the levels of PTG among community workers are unclear. Previous findings suggest that PTG occurs across various populations [9] who experience traumatic events, which has attracted scholars’ interest in exploring its associated variables. Social support [10], deliberate rumination (i.e., a positive cognitive process through which one can reconstruct an existing schema and understanding the cause and meaning of an event) [11,12], and psychosocial interventions [13] have been reported to help people grow after traumatic events. Some studies have identified that self-disclosure [11] and emotional regulation [14] can lead to PTG. More recently, studies have found that empathy helps PTG [4,15]. However, research on PTG among community workers during the COVID-19 pandemic and the mechanisms behind the impact of empathy on their PTG remains relatively limited.

### 1.1. Empathy and PTG

Empathy is a person’s ability to understand the emotions of others (i.e., cognitive empathy) and share emotional states (i.e., affective empathy) [16,17]. Brockhouse et al. [18] regarded empathy as a direct predictor of growth. Several studies provide evidence for this notion, showing that empathy is associated with better adaptive outcomes, such as increased emotional well-being [19], greater social connectedness [20,21], and better health [22]. Through different strategies of empathy that engage one’s cognition (e.g., perspective taking) and affect (e.g., emotion regulation), empathic individuals may be in harmony with their surroundings, which yields beneficial outcomes [22]. One study [15] involving PTG among women with pregnancy loss indicated a significant association between empathy and PTG. Empathy may mediate the relation between PTG and prosocial behavior (i.e., all active actions that benefit others and society, such as caring, cooperating, sharing, volunteering, etc.) [23,24]. Additionally, scholars of PTG have identified empathy as a likely aspect of PTG [25]. Based on a literature analysis, we posited that empathy is positively related to PTG among community workers during the COVID-19 pandemic.

### 1.2. Empathy, Self-Disclosure, Social Support and PTG

Based on the revised PTG model [5], we assumed that self-disclosure mediates the relation between empathy and PTG. Self-disclosure refers to the act of revealing individual information to others, and is the window through which individuals express their feelings and thoughts to others [26]. Self-disclosure, an effective means of social communication, is an important way to reflect individual empathy. Empathic counselors tend to share private and related experiences with clients and to yield more self-disclosure behaviors [27], which can help establish good consulting relationships and improve the consulting effect. Doctors with high empathy can obtain more personal information from patients through self-disclosure, which has a positive impact on diagnosis and treatment [28]. Additionally, empathic individuals’ self-disclosure can promote other people’s expressions of the same emotion or cognition. Empirical studies provide broad support for the association between empathy and self-disclosure. Previous findings have also suggested that the disclosure of traumatic events is associated with health benefits. The therapeutic effects of self-disclosure in patients with PTSD have been identified [29,30]. Moreover, one study involving self-disclosure about a psychotic episode to PTSD, PTG, and recovery following psychosis [31] indicated that the degree of self-disclosure was linked to lower levels of PTSD and higher levels of PTG and recovery. A recent study involving PTSD and growth in adolescents during the COVID-19 pandemic reported that self-disclosure mediates the link between open parent–child communication and PTG [32]. During the disclosure of experiences to others, speakers form a cognitive awareness of their content of disclosure [33]. Indeed, individual self-disclosure involves integrating and constructing their own stressful experiences and emotions and expressing them using structured language. The essence of this process lies in the understanding and construction of individual stressful events [34]. Based on the empirical evidence, we proposed a hypothesis that self-disclosure mediates the effect of empathy on PTG.

Based on the theory of crises and personal growth [35], the social environment is a critical resource that can help one grow following adverse experiences, and social support is an important indicator of the social environment. Social support refers to both psychological and material resources from family members, friends, and colleagues, and is believed to be essential for mental health protection [36,37]. The revised PTG model [5] regards social support as an important predictor of positive changes following traumatic events. For example, both cross-sectional and longitudinal studies have found a positive relation between social support and PTG [37,38] that persists even after accounting for other influencing variables [39]. A longitudinal investigation of the effect of social support on PTG during the COVID-19 pandemic found that human social support significantly predicted PTG, whereas perceived support from pets predicted PTG after a month [40]. By synthesizing the results of 200 studies, a recent meta-analysis revealed a moderately positive correlation between empathy and PTG [41]. Given the repeated finding that social support positively impacts PTG, we hypothesized that social support may foster community workers’ positive psychological changes. In addition, we proposed that empathic individuals obtain more support from others, compared with non-empathic individuals. Empathy is an important component of social interactions, has the value of maintaining interpersonal relationships, and enables one to more positively understand the care or support received from others [42]. Individuals with low empathy may not care about others’ concerns or support for them. These factors may further affect the relationships with others. Previous studies have indicated that empathy enables individuals to have good interpersonal relationships and obtain social support [43]. Recent research has suggested that social support mediates the relation between empathy and prosocial behavior [44]. However, to the best of our knowledge, no studies have examined whether social support mediates the link between empathy and PTG among community workers during the COVID-19 pandemic. Combining the preceding theories and models, we hypothesized that social support mediates the effect of empathy on PTG.

We examined the mediating roles of both self-disclosure and social support in the relation between empathy and PTG. Compared with a simple mediation model, this integrated multiple mediation model is capable of exploring multiple mechanisms from empathy to PTG simultaneously [45]. Therefore, it offers more insights into how empathy is associated with community workers’ PTG during the COVID-19 pandemic, which has important implications for encouraging more PTG. According to the revised model of PTG [5], in narrating one’s personal experience, the emotional aspects and details of the events are revealed, resulting in intimacy with and support from others. Thus, self-disclosure and social support may mediate the relation between empathy and PTG. The available research provides support for the link between self-disclosure and social support. For example, studies on people with mental illness have found that disclosure of mental illness can lead to receiving social support [46]. Further studies have documented that self-disclosure influences psychological well-being through social support as a mediating factor [11]. Given the research supporting the mediator model, we posited that self-disclosure and social support multiply mediate the association between empathy and PTG.

### 1.3. Research Objectives

As discussed, limited attention has been granted to empathy and PTG among community workers. The mental health of these individuals is critical to the current outbreak response. Research focusing on their positive psychological growth is timely and important. To address the aforementioned gap in the literature, our study aimed to identify the relation between empathy and PTG among Chinese community workers and examine the multiple mediating effects of self-disclosure and social support. Therefore, we analyzed the relations among the empathy, self-disclosure, social support, and PTG of Chinese community workers using a multiple mediation model of self-disclosure and social support on empathy and PTG (Figure 1). We expected our analyses to yield meaningful information that can promote Chinese community workers’ PTG during the pandemic, which is crucial for their mental health and outbreak response.

## 2. Methodology

### 2.1. Measures

For empathy, the Chinese version of the Interpersonal Reactivity Index(C-IRI) was used. The IRI is the most widely used measure of empathy [47].The Chinese version was revised by Zhang [48]. It contains 22 items that respondents rated using a five-point Likert scale (0 = does not describe me well, 4 = describes me very well). The tool includes four subscales: perspective-taking (IRI-PT, 5 items), personal distress (IRI-PD, 5 items), empathic concern (IRI-EC, 6 items), and fantasy (IRI-FS, 6 items). The revised Chinese version of this scale has good reliability and validity and is suitable for use as a tool for assessing empathy in the Chinese population [49]. Based on the current sample, the Cronbach’s alpha coefficient was 0.62.

The Distress Disclosure Index (DDI), developed by Kahn and Robert [50], was used to assess participants’ self-disclosure. The questionnaire has 12 items that assess the extent to which individuals share personal information regarding their feelings and worries with others. All items are rated on a five-point Likert-type scale ranging from 1 = strongly disagree to 5 = strongly agree. Our study used the revised DDI [51], which yielded the following values: Cronbach’s alpha coefficient of 0.866, split-half reliability of 0.847, and test–retest reliability of 0.847. The revised scale retains the 12 items of the original scale, among which items 2, 4, 5, 8, 9, and 10 are reverse-scored. We used the total scores for all 12 items to assess the self-disclosure status of individuals, with higher scores indicating higher levels of self-disclosure. We set the following ranges for low-, medium-, and high-level self-disclosure, respectively: 12–29, 30–44, and 45–60. In this study, the Cronbach’s alpha coefficient was 0.82.

Social support was measured using the Social Support Rate Scale (SSRS), which was initially developed for the Chinese population [52]. The SSRS includes 10 items and assesses social support in three dimensions: subjective support (items 1, 3, 4, 5), objective support (items 2, 6, 7), and support utilization (items 8, 9, 10). Items are rated on a four-point Likert scale (1 = none, 2 = slight, 3 = moderate, 4 = great). Item 5 contains choices, and each choice is rated from 1 (never) to 4 (always), generating the total score. The score of items 6 and 7 is the count of the choice. Higher total scores indicate better social support. The SSRS has been widely used in different Chinese populations and proven to have good reliability and validity [53]. In this study, the Cronbach’s alpha coefficient was 0.71.

To measure positive changes during the COVID-19 pandemic among Chinese community workers, we used the Chinese version of the Post-Traumatic Growth Inventory (C-PTGI). The PTGI was developed by Tedeschi and Calhoun [54], and it was modified and validated by Wang et al. [55] for the Chinese population. Tedeschi and Calhoun’s initial scale contains 21 items; however, we used the 20 items arranged by Wang et al. This scale consists of five subscales; however, we used only the total score in the analysis. Each item is rated on a scale from 0 (totally disagree) to 6 (totally agree), with higher total scores indicating higher PTG. The C-PTGI is a popular and widely used measurement in China [36]. In our study, the Cronbach’s alpha coefficient for the 20 items was 0.95.

### 2.2. Participants

G*Power 3.1.9.7 indicated that a minimum sample size of 314 was required to reach a statistical conclusion based on a power of 0.95 (α = 0.05, 1 − β = 0.95) and a medium-small effect size (|r| = 0.2, estimated from the study of Wang et al. [56]). Thus, after making recruitment arrangements with 70 communities in Weifang, Shandong, China, our study invited 520 community workers to complete the online questionnaire survey, which included items on demographics, empathy, self-disclosure, social support, and PTG. After data cleansing (we excluded 106 invalid responses), we obtained a final sample of 414 participants, with a mean age of 39.46 years (SD = 9.04) (Table 1).

Inclusion criteria for participation were as follows: (a) the staff of community party committees, community neighborhood committees, and community service stations, as well as volunteers and community nucleic acid testing personnel who perform community anti-epidemic tasks during the novel coronavirus pneumonia epidemic; (b) adults aged 18 years or older; (c) have the ability to read and understand Chinese questionnaires, and be able to communicate in normal language; (d) volunteer to participate in this survey and sign the informed consent form.

### 2.3. Procedure

Ethical approval was granted by the ethics committees of the first author’s university. Informed consent was obtained from both community workers and their leaders before completing the measures. All participants were informed that their privacy would be protected. The participants completed the self-report questionnaires online during 1 June to 1 July 2022 after the second wave of the outbreak in Weifang, Shandong, China. It took approximately 15 min to finish all items. We recruited 414 Chinese community workers using convenience sampling.

### 2.4. Data Analytic Strategy

Because all the data were generated by self-report, they could have been affected by common method bias, which could, in turn, decrease the validity of the results. Thus, during data collection, we adopted methods to control for common method deviations, such as using screening and reverse questions. For all variables collected using an online self-report method, the Harman single-factor method should be further used to test for common method deviation. Thus, we conducted exploratory factor analysis on all items of the scales used. The results showed that 18 factors with characteristic root greater than 1 were precipitated by non-rotation and rotation, and the variation explained by the first factor obtained by non-rotation and rotation was 16.78% and 13.28%, respectively, which was significantly lower than the 40% critical standard [57]. Thus, our data had no obvious common method bias.

We used IBM SPSS Statistics for Windows 25.0 for descriptive statistical analyses. We checked for the mean and standard deviation and the skewness and kurtosis of the data for parametric statistical analyses. None of the absolute skewness values exceeded 1.5, and none of the absolute kurtosis values exceeded 3.5, indicating that the variances of each variable were close to a normal distribution [58]. For testing the mediating roles of self-disclosure and social support, we used PROCESS Macro 3.5 model 4 [45] separately. PROCESS Macro 3.5 Model 6 [45] was used to test a sequential mediation model of self-disclosure and social support in the relation between empathy and PTG. Finally, bootstrapping with 5000 resamples and 95% confidence intervals was used to analyze the significance of the indirect effects in the mediation model.

Statistical multicollinearity problems occur when the variance inflation factors (VIFs) are greater than 5 or 10 and tolerance is less than 0.1 or 0.2 [59]. Because the tolerances of predictors in our study were in the range of 0.815–0.920 and VIFs were 1.087–1.227, multicollinearity was not a problem. Additionally, the Durbin–Watson statistic was 1.958, which indicated the absence of autocorrelation in the sample. According to the statistical analysis results in Table 1, the current research takes the community workers’ age, educational background, and working years of the subjects as the control variables.

## 3. Results

### 3.1. Relations among the Variables Involved in PTG

Bivariate correlation analysis revealed a significant positive correlation between empathy and PTG (r = 0.42, *p* < 0.01) (Table 2). Thus, the proposed hypothesis was supported. Empathy was also positively correlated with self-disclosure (r = 0.24, *p* < 0.01) and social support (r = 0.40, *p* < 0.01). Additionally, self-disclosure (r = 0.24, *p* < 0.01) and social support (r = 0.46, *p* < 0.01) were positively correlated with PTG. Self-disclosure was positively correlated with social support (r = 0.23, *p* < 0.01).

### 3.2. Mediating Role of Self-Disclosure

We used Model 4 of the PROCESS Macro 3.5 [45] to test the mediation of self-disclosure between empathy and PTG. After controlling for age, educational background, and working years of the subjects, empathy significantly predicted self-disclosure (β = 0.244, *p* < 0.001) and self-disclosure positively predicted PTG (β = 0.457, *p* < 0.001). At the same time, the residual direct effect was also significant (β = 0.94, *p* < 0.001). Therefore, self-disclosure partially mediated the relation between empathy and PTG (indirect effect = 0.11, 95% CI = 0.05 to 0.19). This model accounted for 10.51% of the variance in PTG. Thus, Hypothesis 2 was supported.

### 3.3. Mediating Role of Social Support

After controlling for age, educational background, and working years of the subjects, empathy positively predicted social support (β = 0.44, *p* < 0.001) and social support in turn positively predicted PTG (β = 0.75, *p* < 0.001). Simultaneously, the residual direct effect was significant (β = 0.72, *p* < 0.001). Therefore, social support partially mediated the relation between empathy and PTG (indirect effect = 0.33, 95% CI = 0.22 to 0.45). This model accounted for 31.54% of the variance in the PTG. Thus, our hypothesis was supported.

### 3.4. Verification of the Multiple Mediating Model for PTG

To examine the multiple mediating effect of self-disclosure and social support on empathy and PTG, we used Model 6 of the PROCESS Macro 3.5 [45]. We observed significant pathways from empathy to PTG via self-disclosure (indirect effect = 0.07, 95% CI = 0.01 to 0.14) and via social support (indirect effect = 0.27, 95% CI = 0.18 to 0.38). Therefore, self-disclosure and social support mediated the relation between empathy and PTG. Moreover, the sequential pathway from empathy to PTG via self-disclosure and social support was significant (indirect effect = 0.04, 95% CI = 0.02 to 0.07). Thus, empathy was serially related to higher self-disclosure (β = 0.24, *p* < 0.001, social support (β = 0.39, *p* < 0.001, and PTG (β = 0.67, *p* < 0.001). The residual direct pathway from empathy to PTG was significant (β = 0.67, *p* < 0.001). Thus, self-disclosure and social support partially mediated the link between empathy and PTG. The details are shown in Table 3 and Table 4 and Figure 2.

## 4. Discussion

Given the prolonged nature of the COVID-19 pandemic and its long-term psychological impacts on community workers, assessing the factors that play a role in PTG may foster positive personal growth among community workers. However, previous studies have largely ignored this population. Currently, limited research has investigated PTG among community workers during the COVID-19 pandemic and on the mechanisms behind the impact of empathy on their PTG. To address these gaps, we investigated the association between empathy and community workers’ PTG in the context of Chinese culture. Furthermore, based on the revised PTG theory, we formulated a multiple mediation model to clarify the mechanisms underlying this relation. As expected, self-disclosure and social support mediated the relation between empathy and PTG, both in parallel and sequentially. These results advance the current understanding of the association between empathy and PTG.

The results indicated that self-disclosure is a crucial explanatory mechanism through which empathy relates to community workers’ PTG. Specifically, empathic community workers tended to score higher on self-disclosure, which was related to higher levels of personal growth during the pandemic, consistent with past research [11,28]. In this sense, high self-disclosure is not only a reflection of empathic ability but also an internal antecedent of PTG. This finding reveals the association between empathy and community workers’ PTG. In addition to the overall mediation results, each individual relation was noteworthy. In the first part of the mediation process (i.e., empathy → self-disclosure), our results demonstrated that empathy is positively related to self-disclosure. This finding is congruent with prior research that suggested that empathic individuals are prone to self-disclosure behaviors [28]. Additionally, in daily life, individual empathy includes self-disclosure behavior [60]. In the second part of the mediation process (i.e., self-disclosure → PTG), we found that self-disclosure was positively associated with PTG. This finding supports the revised PTG model in that self-disclosure is essential for promoting individual growth [5]. Indeed, robust evidence confirms a reliable association between self-disclosure and PTG [31,32]. As an effective way of interpersonal communication, self-disclosure allows people to express their needs and emotions, through which individuals may achieve personal growth. Notably, our study revealed that self-disclosure only partially mediated the relation between empathy and PTG, and that the indirect effect was not very strong. This may be because of the complexity of PTG in daily life. Not all forms of PTG are elicited through self-disclosure. Other factors (e.g., social support) may exert a larger effect on PTG, which would override the influence of self-disclosure. Despite the weak mediating effect, self-disclosure is nevertheless an important mechanism linking empathy and PTG, which deserves special attention.

In line with our hypothesis, our results illustrated that social support is another important mechanism through which empathy is related to community workers’ PTG. Specifically, individual empathy is related to high levels of PTG via social support obtained from others. Previous studies have also shown that social support is a critical mediator in the link between individual empathy and personal growth [37,38,44].Each individual link in this mediation model is noteworthy. For the first part of the mediation process (i.e., empathy → social support), our findings were congruent with prior reports that empathic individuals can obtain more support from others relative to non-empathic individuals [44]. One possible explanation is that empathic individuals are more sensitive to the needs and emotions of others and can maintain harmonious interpersonal relationships. High empathy can lead to an active understanding of care and support from others [42]. Meanwhile, individuals with low empathy may not care about others’ support or concern for them. This may further affect their interpersonal relationships and others’ attitudes toward them. Thus, high empathy is conducive to acquiring social support. In the second part of our mediation model (i.e., social support → PTG), we found that high levels of social support were associated with high levels of PTG. This finding supports the theory of thriving through relationships affirming that social support is an important environmental resource for individuals to flourish [61]. Accordingly, many studies have revealed the pivotal role of social support in mental health [4,15,40]. Moreover, social support can help individuals cope successfully with adversities and actively pursue opportunities for growth and development [61]. Remarkably, social support also partially mediated the relation between empathy and PTG, and the model accounted for 31.14% of the variance in PTG. This result is in line with previous research that found that subjective support partially mediated the relation between deliberate rumination and PTG [37]. A positive correlation between social support and PTG has been identified in prior research in diverse populations [4,15]. Therefore, enabling individuals to perceive and obtain more social support is crucial to interventions aimed at individual growth.

Finally, our findings revealed that self-disclosure and social support mediated the relation between empathy and PTG, both in parallel and sequentially, confirming the proposed hypothesis. On the one hand, high empathy is related to increased self-disclosure and social support, both of which are positively associated with PTG [4,31]. On the other hand, the effect of empathy on PTG was sequentially mediated by self-disclosure and social support. Self-disclosure is also positively associated with social support [46]. Individuals with relatively low levels of self-disclosure receive low social support, as others cannot accurately recognize their needs. Therefore, they may be less likely to perceive personal growth.

To sum up, although prior studies have investigated the effects of empathy on PTG [15], few studies, if any, have examined the mechanisms behind the impact of empathy on PTG in the context of the COVID-19 pandemic. Based on the revised PTG model [5], our study illustrated the mediating roles of self-disclosure and social support simultaneously. Our multiple mediation model comprehensively elucidated how empathy is related to community workers’ PTG. In addition, the study also provides some enlightenment for psychological crisis intervention under the epidemic situation. Researchers and mental health workers need to pay attention to the empathy, self-disclosure, and social support of community workers when understanding and intervening with PTG. On the one hand, we can improve the empathy of community workers to help them reveal their emotional distress and strengthen their positive cognition. On the other hand, social support is the external resources to promote the growth of individuals. Individuals can be trained to obtain the skills of such external resources and strengthen the support of units and families for community workers.

## 5. Limitations and Future Directions

Some limitations of this study must be acknowledged. First, although the mediation model we used was based on empirical studies and theoretical foundations, our sample was not representative of the entire Chinese community worker population because the data were collected using convenience sampling. Therefore, further studies are necessary to verify the relations revealed. Second, the cross-sectional design of the study makes it impossible to infer causality. Further longitudinal studies and intervention experiments are needed to examine the validity of the multiple mediation model. Finally, we collected data mainly through self-report measures of community workers. Although extensively used, self-report measures may generate bias. For instance, the self-report measure of PTGI requires respondents to engage in a mentally taxing procedure based on the assumption that individuals can think of previous trait levels accurately. However, perceptions of change are not always linked to actual changes, potentially leading to incorrect reports of PTG. The use of informant reports would be ideal for future research.

## 6. Conclusions

In summary, using a sample of Chinese community workers, our study takes an important step in examining the multiple mediation model through which empathy is associated with PTG. Our findings showed that self-disclosure and social support mediate the relation between empathy and PTG, both in parallel and sequentially. A better understanding of how empathy is related to PTG can help improve community workers’ PTG amid the ongoing COVID-19 pandemic.

## Figures and Tables

**Figure 1 ijerph-19-15739-f001:**
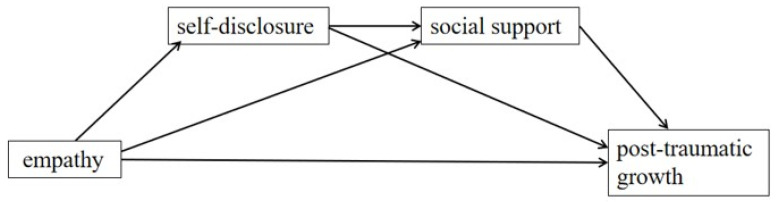
Proposed multiple mediating model.

**Figure 2 ijerph-19-15739-f002:**
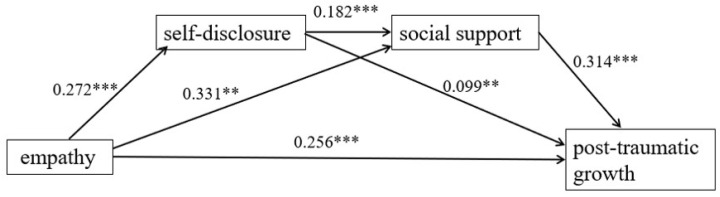
Mediating model of self-disclosure and social support on empathy and post-traumatic growth (standardized coefficients; ** *p* < 0.01, *** *p* < 0.0001).

**Table 1 ijerph-19-15739-t001:** Baseline characteristics and differences in PTG score of community workers (*N* = 414).

Variables	Frequency (Percentage)	PTG Score (M ± SD)	*t/F*	*p*
Age (years)			5.379 ***	<0.001
≥20–29	63 (15.3)	(65.11 ± 14.30)		
≥30–39	149 (36.0)	(57.76 ± 20.59)		
≥40–49	135 (32.6)	(64.84 ± 18.12)		
≥50–59	59 (14.3)	(68.46 ± 12.71)		
≥60	6 (1.4)	(55.33 ± 19.16)		
Sex			−0.355	0.723
Male	128 (30.9)	(62.25 ± 20.47)		
Female	286 (69.1)	(62.94 ± 17.22)		
Marital status			1.629	0.197
Unmarried	46 (11.1)	(63.17 ± 16.27)		
Married	360 (87.0)	(62.42 ± 18.58)		
Separated/Widowed	8 (1.9)	(74.13 ± 10.99)		
Educational background			2.810 *	0.039
High school and below	126 (30.4)	(65.49 ± 16.37)		
Junior college	141 (34.1)	(63.62 ± 17.39)		
Undergraduate	139 (33.6)	(59.76 ± 20.01)		
Master	8 (1.9)	(55.00 ± 23.37)		
Working years			2.963 *	0.020
≤3 years	65 (15.7)	(64.82 ± 16.07)		
3–5 years	36 (8.7)	(67.33 ± 14.59)		
5–10 years	71 (17.1)	(56.65 ± 20.36)		
10–15 years	101 (24.4)	(62.38 ± 20.87)		
≥16 years	141 (34.1)	(63.90 ± 16.32)		

* *p* < 0.05, *** *p* < 0.001.

**Table 2 ijerph-19-15739-t002:** Correlational matrix of empathy, self-disclosure, social support, and post-traumatic growth (*N* = 414).

Variables	①	②	③	④
① Empathy	1			
② Self-disclosure	0.24 **	1		
③ Social support	0.40 **	0.23 **	1	
④ Post-traumatic growth	0.42 **	0.24 **	0.46 **	1
*M*	51.57	38.06	44.38	62.73
*SD*	6.97	6.25	8.13	18.26
Skewness	−0.06	0.49	−0.41	−0.95
Kurtosis	0.28	1.25	−0.24	1.23

** *p* < 0.01.

**Table 3 ijerph-19-15739-t003:** Regression analysis of the relationship between variables. (*N* = 414).

Outcome Variable	Predictor Variable	*β*	*t*	LLCI	ULCI	*p*	*R* ^2^	*F*
Self-disclosure	Empathy	0.244	5.676	0.159	0.328	<0.001	0.118	4.477
Social support	Empathy	0.385	7.130	0.279	0.492	<0.001	0.237	9.573
	Self-disclosure	0.237	3.918	0.118	0.356	<0.001		
PTG	Empathy	0.671	5.554	0.433	0.908	<0.001	0.331	14.128
	Self-disclosure	0.289	2.229	0.034	0.545	<0.05		
	Social support	0.705	6.697	0.498	0.912	<0.001		
PTG	Empathy	1.054	9.019	0.824	1.284	<0.001	0.235	10.251

**Table 4 ijerph-19-15739-t004:** The chain mediating effect of self-disclosure and social support.

Section	Effect	SE	LLCI	ULCI
Total	0.383	0.061	0.268	0.508
EM → SD → PTG	0.071	0.031	0.012	0.135
EM → SS → PTG	0.272	0.053	0.175	0.381
EM → SD → SS → PTG	0.041	0.014	0.018	0.071

Note: EM = empathy; SD = self-disclosure; SS = social support; PTG = post-traumatic growth.

## Data Availability

The raw data supporting the conclusions of this article will be made available by the authors without undue reservation.

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
