# Peer review of "Empathy and Post-Traumatic Growth among Chinese Community Workers during the COVID-19 Pandemic: Roles of Self-Disclosure and Social Support"

_ijerph, 2022, doi:10.3390/ijerph192315739_

Round 1
Reviewer 1 Report
Interesting paper, which adds a useful insight into the effects of the pandemic. Especially the focus on self-managing trauma. Useful to explain the difference between community workers and other workers. Also, technical terms like deliberate rumination, prosocial etc need to be explained.
Author Response
Dear Reviewer, Thank you very much for reviewing this manuscript. We have prepared a revised submission to address all concerns and recommendations. We appreciate the opportunity to revise this work.

Reviewer 2 Report
The authors made an excellent overview of previous research on their analyzed dimensions. The research methodology is clear and valid. Research results are presented sufficiently, and their discussion has been made. Conclusions support the findings, and this article could interest practice and scientists. I recommend publishing it after a minor English spell check.
I think it is really well organized and written and provides scientifically and practically relevant research results. The authors have analyzed and systemized the newest research, prepared and validated suitable research methodology, clearly analyzed research data, and discussed them critically overviewing their work in comparison with research in the past and needs for the future.
The only thing that could be discussed and overthought is structure. I would describe the research object as the first thing in the article and would make it a part of the introduction. I would explain measures first in the methodology and just later would write about sample size and respondents. Those things are not essential and could depend on cognitive perception. Important thing is that all those critical aspects are described and validated.
Author Response
Dear reviewer,
Thank you very much for your kindly review this manuscript. We have prepared the revised submission and aimed at addressing all the concerns and recommendations. We appreciate the opportunity to revise this work.

Reviewer 3 Report
Very interesting. It would be appropriate to explain a little more:
1. The process of recruiting the population sample that has responded. How much time in total did they spend on the survey? when did it start and when did it end? post-pandemic? where? how?...
2. Explain a little more about the measuring instruments. Cite authors who have used them and justify why they are and not others.
3. Why have men responded more than women? What are community workers? can you specify more?
4. Explain a little more the practical usefulness of the results obtained. Precisely what are they for?
5. How have you obtained the consent of the participants? Have they signed any documents? And how have they guaranteed the anonymity of the participants?
Author Response

(The authors gave the same response as above.)
